# Yoko–Dovyren Layered Massif: Composition, Mineralization, Overburden and Dump Rock Utilization

**Evgeniy V. Kislov** [1,]* and **Lyudmila I. Khudyakova** [2]

[1]  Geological Institute, Siberian Branch, Russian Academy of Sciences, 670047 Ulan-Ude, Russia
[2]  Baikal Institute of Nature Management, Siberian Branch, Russian Academy of Sciences,
     670047 Ulan-Ude, Russia; lkhud@binm.ru
*   Correspondence: evg-kislov@ya.ru

**Abstract:** Ultramafic–mafic complexes are widely developed in the Earth's crust. They contain deposits of various minerals. The Yoko–Dovyren intrusive in the North Baikal Region, Russia, is considered an example of an intrusive containing diverse mineralization: Ni-Cu, Platinum group elements, Cr, Zr, B, and blue diopside. During the development of the deposit, a huge amount of magnesium-containing rocks are moved to dumps and have a negative impact on the environment. To minimize this process, overburden and host rocks need to be involved in production, thereby avoiding the movement of rocks into dumps. The construction materials production is main industry using this rocks. Therefore, the purpose of these studies was to determine the quality of magnesium-containing rocks and the possibility of their use in construction. As a result of the complex works performed, it has been determined that these rocks have required physical and mechanical characteristics. Concretes in large and small aggregates from magnesium-containing rocks were obtained. It has been concluded that they are superior to concrete from granite rubble and quartz sand in terms of their strength indicators. The use of magnesium-containing rocks, without allowing them to fall into dumps, will allow us to create clean, environmentally safe mining enterprises.

**Keywords:** dunite–troctolite–gabbro massif; Ni-Cu; PGE; chromitite; magnesium rocks; building materials

## 1. Introduction

Ultramafic–mafic complexes are found everywhere in the world. They include various types of mineral deposits: Ni-Cu and PGE [1], Cr [2], Fe-Ti-V [3], asbestos [4], nephrite and jadeite [5], talc, magnesite, vermiculate [6] and others. Kimberlite and lamproite pipes contain diamonds [7,8].

In their natural state, the complexes do not have any impact on the environment. However, during development, they become a source of negative influence on nature. All these deposits contain a small proportion of useful components. This is especially true for diamonds and metal ore deposits. More than 90% of the extracted rock mass goes to dumps, considering dilution during mining. Both mining itself and its consequences are negative. Moreover, the load on ecosystems continues for a long time [9,10], it is global and represents a serious problem for all types of ecosystems [11].

During the operation of the mining enterprise, the landscape of territories is changed, the land plots are withdrawn for waste rocks. The removal of soil cover, deforestation, the destruction of vegetation, the contamination of soil and water resources, and the loss of biodiversity in a mining zone change the environment on a local and regional scale [12–14].

The destruction of mountain peaks by blasting operations changes the hydrological regime of river flows and water quality [15–17]. The developed spaces are filled with water, turning into reservoirs that

differ from natural lakes [18]. Many of them are toxic, which poses a threat to adjacent ecosystems [19], although some of them are used for entertainment purposes [20].

Among the main problems of mining enterprises are dumps and tailings, which pose a great risk to the environment and people. When developing deposits, only a small part of the extracted rock mass is used. Overburden and host rocks, the amounts of which exceed 90% of the extracted ore, as well as enrichment waste, are stored in dumps. The dumps occupy large areas near the developed fields, and their impact on the environment is of a transboundary nature [11,21]. However, dump rocks should be considered not only as sources of pollution, but also as potential mineral resources [22,23]. This will allow us to achieve the rational use of natural resources, maintaining a balance between the extraction of mineral resources and environmental preservation [24–26].

The use of waste rocks in the production of building materials allows us to dispose of waste, prevent contamination of the area and its occupation in dumps, save money in the delivery of construction materials, and reduce carbon dioxide emissions in cement production. However, magnesium-containing rocks that are part of the ultramafic–mafic complexes are not used in the production of construction materials, and remain in dumps due to the biased attitude towards them. Therefore, the issues of their disposal are relevant and require further study.

The solution to this problem is shown in the example of the Yoko–Dovyren layered dunite–troctolite–gabbro massif of the North Baikal region, Russia.

## 2. Yoko–Dovyren Massif

### 2.1. Location, Structure and Composition of the Yoko–Dovyren Massif

The Yoko–Dovyren intrusion is located in a folded frame of the South Siberian craton in the North Baikal region, Russia, 60 km northeast of Lake Baikal. It lies sub-concordant to the carbonate–terrigenous rocks (mainly black shale) of the Synnyr rift [27–29].

The Yoko–Dovyren massif is $26.0 \times 3.5$ km$^2$ in size. It is a part of the Synnyr–Dovyren volcano–plutonic complex. The complex also includes underlying plagioperidotite sills and leucocratic gabbronorite dikes [28,30], with similar dikes in the roof. The complex consists of effusives of the Synnyr ridge overlapping these bodies (Figure 1), from high-Ti basalts of the Inyapuk suite low-Ti andesites and basalts of the Synnyr suite [31]. Geochemical and isotopic data indicate a genetic relationship between the intrusive and low-Ti volcanic rocks [32].

The U-Pb ages using zircons are 728.4 ± 3.4 Ma for the Yoko–Dovyren massif and 722 ± 7 Ma-for the associated volcanics [33,34]. Using the baddeleytte from pegmatoid gabbronorite in the roof, the U-Pb dating gave 724.7 ± 2.5 Ma [35].

As a result of tectonic movements, the Yoko–Dovyren massif has an almost vertical orientation. The intrusion is composed of contact rocks (quenched gabbronorites and picrodolerites, above-plagioclase lherzolites), which turn into the main stratigraphic sequences of five zones (from bottom to top): dunite (Ol + Chr) → troctolite + plagiodunite (Ol + Pl + Chr) → troctolite + olivine gabbro (Ol + Pl + Chr ± Cpx) → olivine gabbro (Pl + Ol + Cpx ± Chr) → olivine gabbronorite (Pl + Ol + Cpx ± Opx) → quartz gabbronorites and pigeonite-containing gabbro (Pl + Cpx ± Opx ± Pig). The quartz gabbronorites and pigeonite-containing gabbros in the roof part belong to an additional intrusion synchronously with dikes of gabbronorites.

At the base of the Yoko–Dovyren massif, below the quenched contacts, one can trace extended (up to several km) plagioperidotite-intrusive sills [28,36]. The results of the geological mapping of the massif show (Figure 1) that in the central, thickest part, these extended intrusive sills can occur with the basal base of the intrusion, probably with a somewhat deformed cone shape [30,36].

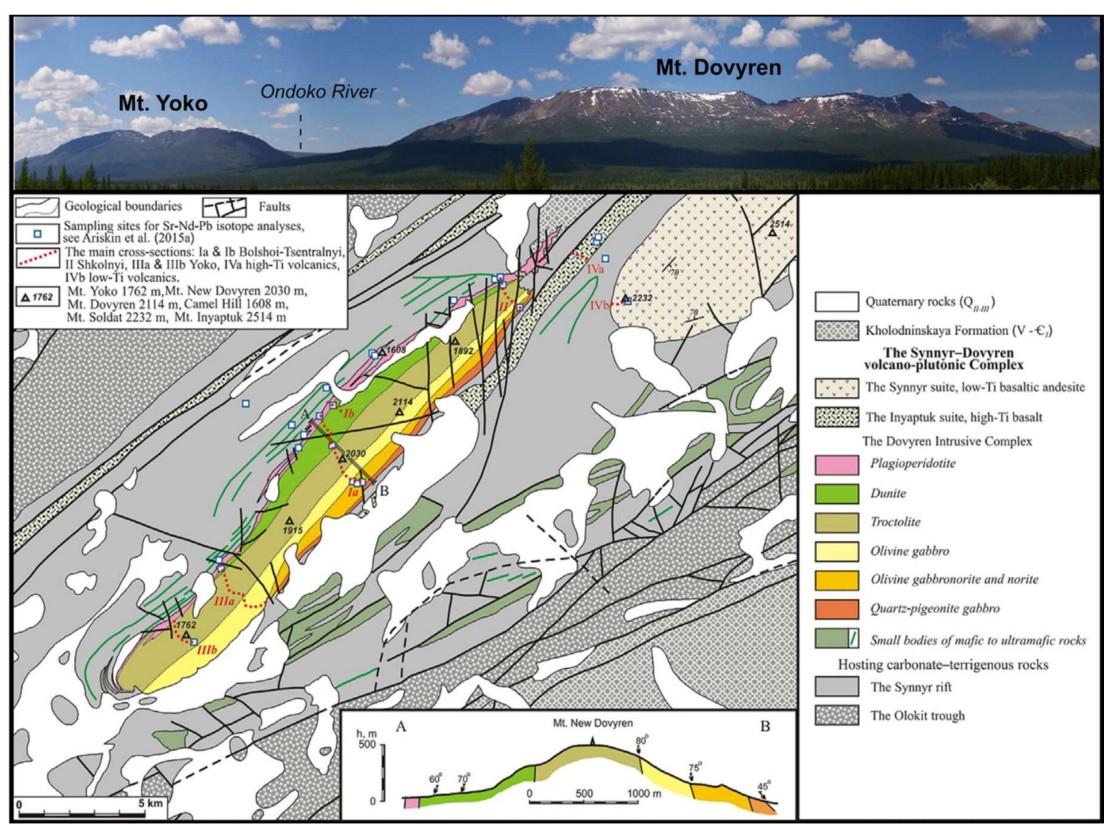

**Figure 1.** A photograph of the Yoko–Dovyren massif facing the northwest (top) and a schematic geological map of the southwestern termination of the Olokit trough [30].

## 2.2. Mineralization of the Yoko–Dovyren Massif

### 2.2.1. Sulfide Nickel–Copper Ores

Nickel–copper mineralization, down to net-textured and massive ores, is concentrated along the intrusion's lower contact—the Baikal deposit [27,28,37]. The mineralization was discovered in 1949. The exploration work was intensively carried out in 1959–1963 and, to a lesser extent, in 1976–1979 and 1986–1993.

The richest mineralization is concentrated in plagioperidotites and sills of the same composition extending into the bottom rocks. In plagioperidotites, disseminated and massive sulfide ores are distributed unevenly. The disseminated mineralization (Figure 2) is distributed much more widely than the massive one. The bodies of disseminated sulfide ores can be traced along the strike to 1400–1700 m with an emergence width of 8–25 m (up to 80 m). The orientation of the disseminated mineralization lenses, as a rule, coincides with the strike and the dip of the bottom horizon of plagioperidotites. The bodies of massive ores are usually embedded in areas of sulfide impregnation.

The mineral composition of sulfide ores of the Baikal deposit is represented by the "triad" typical of magmatic sulfide–nickel deposits: pyrrhotite, pentlandite and chalcopyrite; oxides such as magnetite and chromite are found in lesser amounts. Pyrrhotite predominates quantitatively.

The richest ores are known to be at the northeast end of the massif (Ozernyi prospect). The area of disseminated sulfide ores can be traced along the bottom of the intrusion and along the strike and up to 700 m with an average thickness of 8 m. Within this zone, there are a number of veins and lenses of massive sulfide ores. The largest sulfide vein is traceable along the strike to 650 m with a thickness of 0.7–1.0 m. There are also smaller sulfide veins confined to the tectonic zones of the sub-latitude and sub-meridian directions. Their thickness ranges from 0.2 to 1.5 m. Sub-latitude veins are more

extended than sub-meridian veins (15–50 m). According to the drilling data, the veins fall almost vertically (50–70) and are found at depths of more than 500 m.

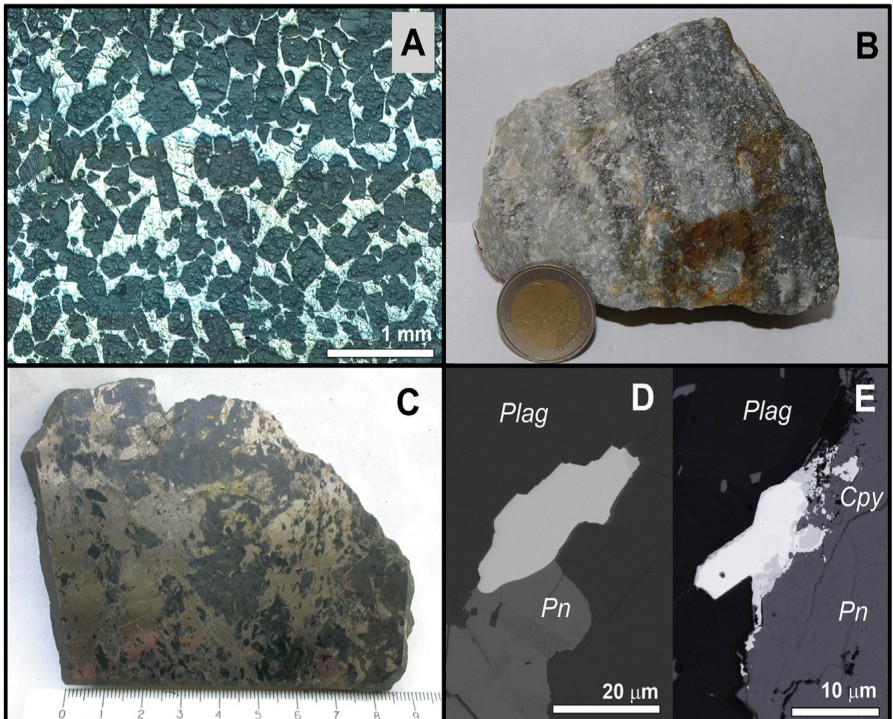

**Figure 2.** Sulfide mineralization of the Dovyren intrusive complex [30]. (**A**) Net-textured Ni-Cu sulfide ore from a sill (sample 07DV107-1); (**B**) low-mineralized PGE-rich anorthosite from the Main PGE Reef (07DV146-1); (**C**) a massive Po-rich sulfide ore from the Ozernyi prospect at the NE termination of the Yoko–Dovyren massif (sample provided by D.A. Orsoev); (**D,E**) back-scattered electron (BSE) images of Pt-Pd-Ag minerals associated with sulfides: (**D**) moncheite in a thin anorthositic vein (the Main PGE Reef, sample 07DV146-2); (**E**) a composite grain of moncheite (light) and telargpalite (grey) in a low-mineralized troctolite. Plag—plagioclase, Cpy—chalcopyrite, Pn—pentlandite.

The disseminated ores are characterized, on the whole, by low concentrations of sulfides (5–10 vol.%), but among them there are areas of densely disseminated and net-textured ores (Figure 2) with higher contents (up to 30%). In such areas, the concentrations of useful components reach industrial levels (0.30–1.05 wt.% Ni). A distinctive feature of the ores of the Yoko–Dovyren massif is their high content of cobalt, which isomorphically enters pentlandite, violarite, nickelin and gersdorfite (up to 12% Co is found in the last two minerals), and forms independent minerals: cobaltite and cobalt mackinawite. In addition to the main components, nickel, copper, cobalt and sulfide ores have increased concentrations of PGE (Pt—up to 0.5 ppm; Pd—up to 3.8 ppm; Rh—up to 0.24 ppm); Au (up to 0.32 ppm); Ag (up to 16 ppm); Se (up to 23 ppm); Te (up to 14 ppm).

The resources of the deposit are estimated to contain 147 thousand tons of nickel, 51 thousand tons of copper and 9.5 thousand tons of cobalt. By drilling to a depth of only 300 m, the deposit has not been studied enough; only one well has been drilled to 750 m.

### 2.2.2. Low-Sulfide Platinum Mineralization of the Main Reef

Much later, layered horizons ("reefs") of petrographic heterogeneous rocks with a low sulfide mineralization of PGE were found in the Yoko–Dovyren massif [28,38–44]. The first, horizon with maximum PGE concentrations, located lower in cross section, is called the Main Reef or Reef I (Figure 1). It is confined to the transition zone from the rhythmically layered troctolite and olivine gabbro series to the massive olivine gabbro.

The Main Reef occupies a varying position from 170 to 280 m above the marker horizon of poikilitic plagiowehrlites. The Main Reef is traced by route intersections for more than 20 km along the strike and, using the terrain, to a depth of about 1 km. A characteristic feature of the Main Reef is schlieren and lenticular segregations of anorthosites (Figure 2), and to a lesser extent of gabbro pegmatites, taxitic troctolites and olivine leucogabbronorites with a thickness from a few cm to a meter or more. Along the strike, they extend sub-concordantly with the layering for 2–5 m, and rarely for 40 or more meters, forming discontinuous en echelon ore zones. Often, they are oriented sub-horizontally, that is, they cut across a sub-vertical layered series.

In ore-bearing pegmatoid anorthosites, the size of olivine crystals is up to 6 mm, bytownite–anorthite up to 12 mm, poikilocrystals of diopside–augite up to $120 \times 40$ mm$^2$, and bronzite up to 50 mm. Thin sulfide impregnation in anorthosites is devoted to areas and bands of Fe-Mg minerals, which are actively replaced by sulfides. Usually, sulfides are less than 1%, but their content can reach 7%. Net textured sulfides are represented by crystallization products of Ni-Cu-Fe-S liquid. Small crystals and their Mss aggregates are predominant and are transformed into intergrowths of troilite, Fe–pyrrhotite, and pentlandite. Small crystals of Iss1, undersaturated with sulfur, transformed into intergrowths of cubanite, troilite and pentlandite, are widespread. Less common are the copperier Iss2, Iss3, Iss4, and Iss5 and solid-state transformation products such as Iss5 (talnakhite). Chalcopyrite, cubanite and talnakhite contain pentlandite exsolution lamellae [43].

PGE minerals are found not only in sulfide aggregates, but also in contact with silicates, as well as in a silicate matrix in close intergrowth with OH-bearing minerals. They are mainly represented by moncheite (Figure 2), tetraferroplatinum, potarite, kotulskite, moncheite and zvyagintsevite, as well as telargpalite, insisvaite, michenerite, paolovite, sperrylite, froodite, mertieite, niggliite, atokite, sobolevskite, majakite, hessite, rustenburgite, stannopalladinite, taymyrite, heversite, palladium germanite, minerals of the gold–silver series, silver amalgam and altaite [28,30,43,44].

The contents of precious metals in rocks are distributed extremely unevenly. Anorthosites are the most enriched, in separate samples of which the precious metals contents reach P-4.1, P-7.8 and A-3.2 ppm. The contents reach: Cu 0.7%, and Ni 0.4%. Hypothetical resources: P-42.4 t, P-66 t.

### 2.2.3. Other Sulfide Mineralization

An impregnation of sulfides was also found on other stratigraphic horizons of the massif; their industrial significance has not yet been determined.

Low-sulfide mineralization Os-Ir-Ru with disseminated Mss-type sulfides occurs in the horizon of plagiodunites lying between plagiolherzolites and dunites [30]. The disseminated mineralization of troilite and pentlandite and, less often, vein-disseminated mineralization with a high content of Pd, occurs in the upper part of the dunite zone in the areas around dolomite xenoliths (the Bolshoy creek, [28]). Low-sulfide high-copper mineralization of PGE occurs in troctolites of the lower part of the layered series—the "Konnikov zone" [30,45].

Sulfide mineralization is in troctolites and vein diopsidites in the lower part of the plagiodunite–troctolite subzone in the areas around the after-dolomite and after-aleurolite xenoliths (the Belyi creek, [28]).

Low-sulfide mineralization occurs in troctolites containing cobalt pentlandite ("under trigger point"). Low-sulfide PGE veins and schlieren of anorthosites and gabbro-pegmatites are found among massive gabbros and gabbronorites [28].

A horizon of relatively sulfide-rich (in places up to ~10 wt.%) olivine-free gabbronorites can be found within the roof zone [28,30,46].

Gabbronorite sills at the roof and bottom: sulfide mineralization is observed in gabbro–pegmatites; quartz–carbonate veins are found to impregnate gabbronorites with quartz; a large impregnation of chalcopyrite and galena is observed in quartz–carbonate veins that does not cause the silicification of gabbronorites, with sharp contacts [28].

Thin pyrite–pyrrhotine impregnation prior to vein mineralization is characteristic of exocontact terrigenous rocks. In addition to pyrite, sphalerite and galena occur in carbonate deposits [28].

### 2.2.4. Chromitites

Wehrlites, diopsidites and chromitites are related to the zone of magnesian skarns at the left board of the upper part of the Bol'shoy (Big) stream (Figure 3). The chromitites are most often represented as a schlier-like stratification of chrome–spinel (Figure 4). Its dimensions are up to 0.5–1 m in length and 10–20 cm in width. The chromitites consist of a 40–60% euhedral chrome–spinel. Veins of massive chromitite with widths up to 2 cm are less frequent. Olivine, grassy green clinopyroxene, chlorite, green or greenish-brown garnet are typical of the chromitites. Clinopyroxene and chlorite often surround chrome–spinel grains.

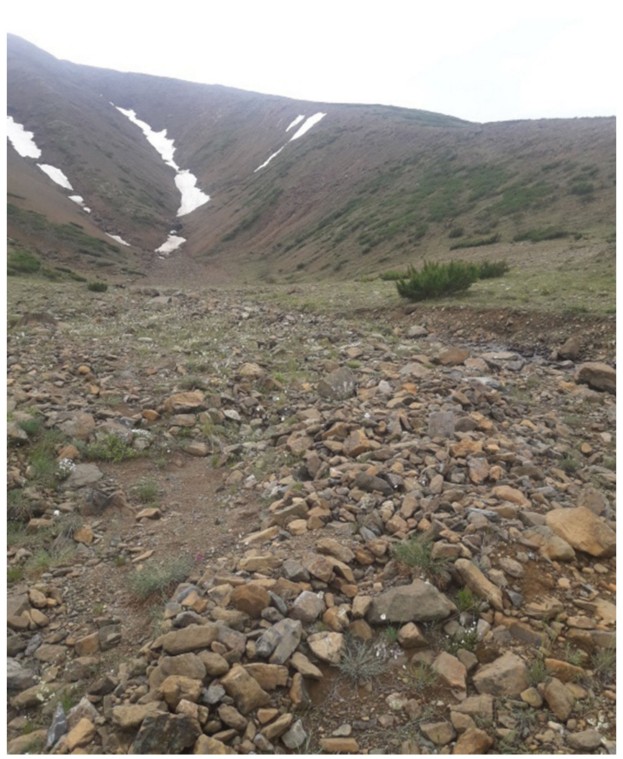

**Figure 3.** Dunite zone: chromitite and wehrlite outcrops are located above the right snow fields.

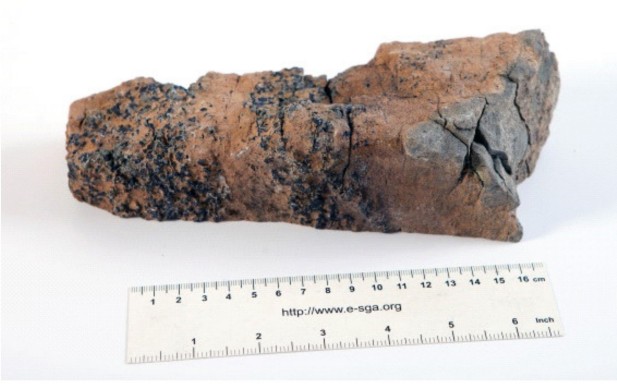

**Figure 4.** Sample B1b: two chrome–spinel layers are parallel to a magnesian skarn.

The chrome–spinel in the chromitites contains picotite and chromepikotite with a prevalence of Al over Cr (Figure 5). Olivine in the chromitites is characterized by higher magnesium (Mg# = 0.97–0.89) in comparison to the olivine of non-contaminated dunites (Mg# = 0.87–0.85) and by a several times higher content of CaO (up to 1.2%, on average 1%). Clinopyroxene of a bright grassy green color is characterized by a high content of $Al_2O_3$ (6–8%), $Cr_2O_3$ (1%) and $TiO_2$ (0.5–0.7%) in comparison to the clinopyroxene of non-contaminated dunites [30,47,48].

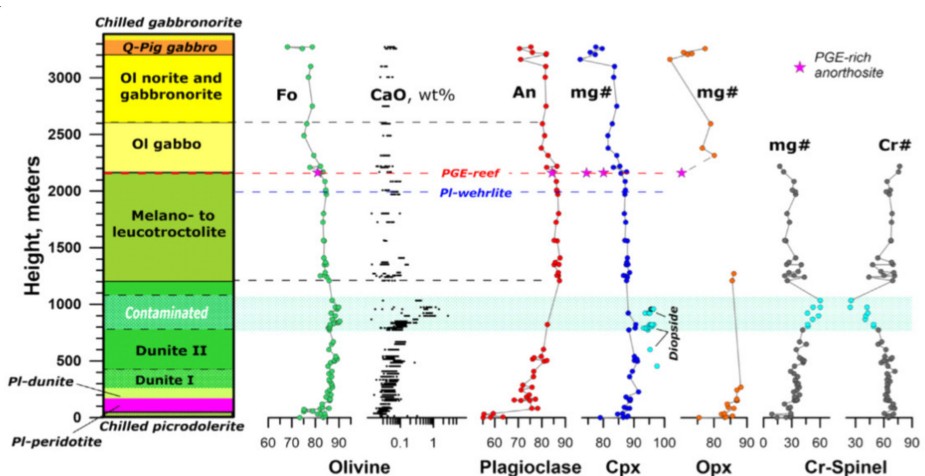

**Figure 5.** Variations in mineral compositions along the Bolshoi-Tsentralnyi cross-section. Compositions of contaminated dunite minerals are highlighted in green. mg# = MgO/(MgO + FeO), Cr# = Cr/(Cr + Al) [30].

Olivine is greenish-brown on a fresh chip and yellow on a weathered crust. It is presented by isomeric euhedral grains from 0.2 mm to 4–5 mm in size. It has crystallographic outlines, mainly in the bulk. Inclusions of roundish olivine grains are also observed in larger grains of olivine, diopside and chrome–spinel. The pronounced parting appearing as a perfect cleavage is typical. It is determined by the microscopic lamels of the monticellite created by the exsolution of high-Ca olivine [49]. Serpentine, chlorite, calcite, and magnetite were formed in numerous cracks. The composition variations in one grain are typical of olivine. A spotted distribution of different compositions is more frequent. Zoning with an increase in Fe content to the periphery is rare. Inclusions of chrome–spinel, magnetite (occasionally with diopside inclusions), monticellite (roundish, facetted or lath-shaped), diopside (roundish or lath-shaped), chlorapatite, pyrrhotine, halite, problematic chlorides of calcium, magnesium, iron, barium and potassium are observed in the olivine [50].

Bright green diopside forms fringes of chrome–spinel crystals and poikilocrists including and, perhaps, corroding roundish grains of olivine. Grossular inclusions (both separate grains, and streaks), magnetite, chrome–magnetite, chrome–spinel, chlorite and chlorapatite are observed in diopside. Then melilite occasionally surrounds the diopside.

Black chrome–spinel forms large isometric grains from 1 to 3 mm in size. It has an isotropic black color with a reddish shade. The majority of the grain content in cracks is filled with serpentine, chlorite, calcite and magnetite.

These sites with different compositions are peculiar for chrome–spinel grains. The chrome-spinel differ in composition from the spinel, with Zn and without Cr to pure chromite. Chrome–magnetite fringes, veins and chains of grains are also typical.

Chlorite, diopside, olivine, magnetite, titanomagnetite, chrome–magnetite, low-aluminous chrome–spinel, grossular, pargasite, phlogopite, apatite, chlorapatite, fluorapatite, vesuvianite (zoning sometimes), nepheline, cuspidine, halite, pentlandite, calcite (iron and rare magnesium content), serpentine, melilite, perovskite, chalcopyrite, djerfisherite and galena form inclusions in chrome–spinel. Nepheline grains are observed as roundish inclusions in diopside. Diopside acquires

grossular and is occasionally consistently chlorite in inclusions (Figure 6). Chrome–magnetite is occasionally formed directly on the border with the inclusion; chromite is less common.

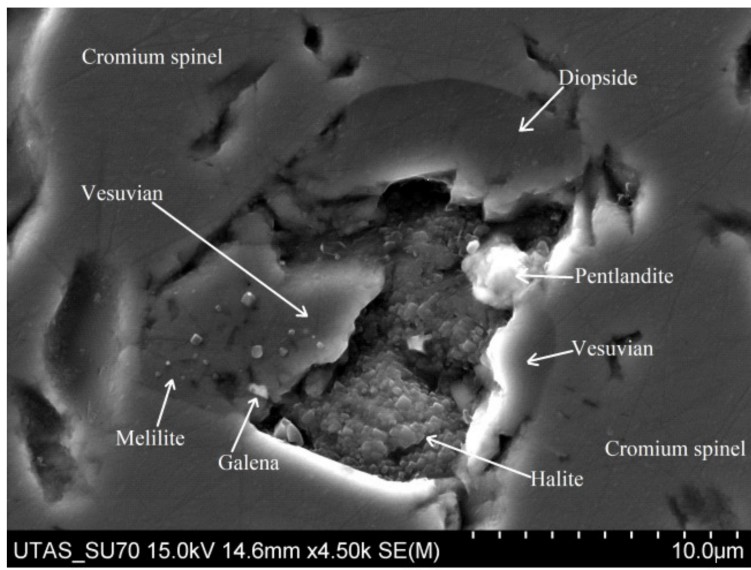

**Figure 6.** An inclusion in chrome–spinel, samples 14–15. Scale is 10 μm.

Chlorite, grossular (with hydrogrossular or occasionally chrome–magnetite in the cores of the crystals), phlogopite, chlorapatite, fluorapatite, calcite, magnetite, pentlandite, chalcopyrite, pyrrhotine, galena and magnesian siderite are also observed in the bulk of chromitite. A streak consisting of diopside, grossular, apatite, chrome–magnetite and chlorite among the larger olivine grains is observed.

The calcic mica clintonite $CaAlMg_2(SiAl_3)O_{10}(OH)_2$ characteristic of skarn is also found. In one case, it forms an inclusion in chlorapatite (sample B1b). In another case, it is on contact between chrome–spinel and diopside (samples 15–15).

Grains of sulfides sometimes contain inclusions of olivine or magnetite. Chlorite laths penetrating anhedral grains of sulfides in all directions are observed. A pentlandite grain contains an inclusion of galena. Zoning crystals with chrome–magnetite in the core and with magnetite in the external zone are met. A pentlandite–calcite–magnetite aggregate in conjunction with chrome–spinel crystals is observed.

Sulfides are replaced by magnetite or chrome–magnetite, and those, in turn, are often replaced by goethite. Bornite, cuprite, Fe oxides, hydroxides, carbonates with Cu, Fe oxides with Si and Mg are all produced on primary sulfides.

Serpentine, calcite, and chlorite fill the cracks in olivine, chrome–spinel, diopside. Magnetite is common in such streaks. Galena and cubanite are found less often. Streaks of white calcite are larger—about 2.0–2.5-mm long and 0.1–0.2-mm wide. Calcite and chlorite fill the interstices between grains of olivine, they also compose euhedral grains. Pyrrhotine–pentlandite and pentlandite–magnetite grains are found in euhedral grains of calcite.

Our research proves that chromitites in contaminated dunites of the Yoko–Dovyren massif represent a high-chromium skarn of the magmatic stage. Its formation is connected to the reaction of picrobasalt melt with $CO_2$ fluid and excess calcium extracted by the decarbonatization of dolomitic xenoliths [50].

### 2.2.5. Boron Mineralization

Axinite–quartz veins with carbonate and pyrite 10–20-cm thick, and series of veins 1–2-cm thick, were found in a diabase dyke 7–8-m thick, which is exposed behind the northwestern wedge of the massif. The dike is crossed by transverse axinite–quartz veins 10–20-cm thick, and series of veins

1–2-cm thick. Wedge-shaped crystals of axinite (the first millimeters—1.5 cm) form brushes (sometimes with radial ray intergrowths) in the selvages of the veins. Individual crystals and fine-grained aggregates of axinite in quartz are characteristic. The crystals are characterized by rough combination striations. Axinite is transparent and clove brown; when weathered, it becomes grayish-beige and opaque. In terms of its chemical composition, it is magnesian–ferruginous axinite with a high content of magnesium and manganese. In quartz aggregates, finely scaled chlorite, acicular to elongated prismatic crystals (also in axinite) and pyrite hexahedra (almost completely replaced by iron hydroxides) are observed in quartz aggregates. Judging by the negative pseudomorphoses, calcite rhombohedra were present in the selvages of the veins [51].

### 2.2.6. Blue Diopside

The development site of the blue diopside—the Snezhniy prospect—is found at the source of the Belyi creek. The mineral is found in a xenolith composed of brucite marbles, in troctolites. The xenolith is 100-m long along the strike and 10–15-m thick (Figure 7).

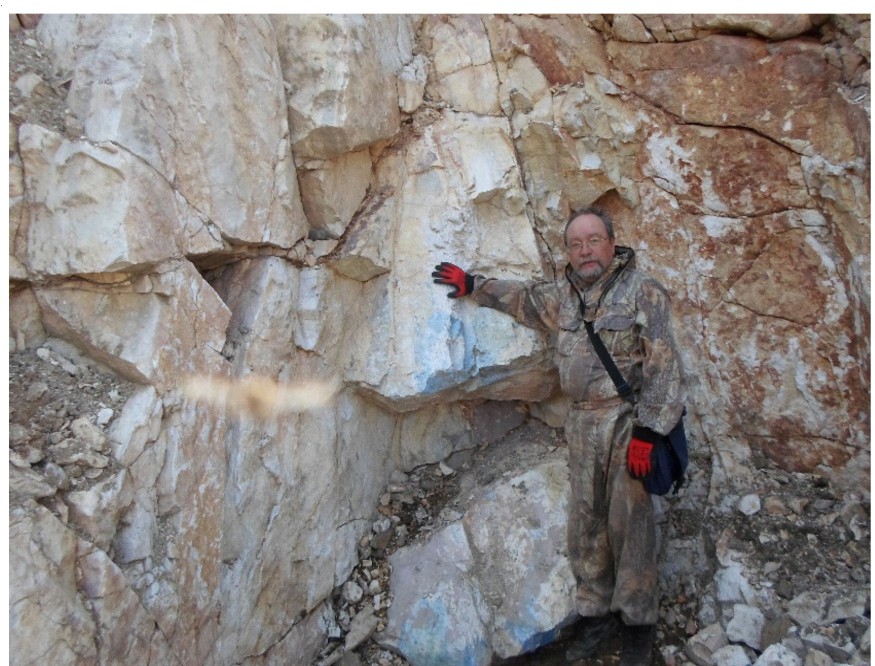

**Figure 7.** Outcrop of blue diopside at the Yoko–Dovyren massif with E.V. Kislov.

Diopside is developed around the lenticular and pipe-like segregations of quartz. Lime–magnesian skarn is developed around them: quartz–diopside + wollastonite + calcite–light porcelaneous monticellite + wollastonite + calcium hydrosilicate–brucite marble. In contact with quartz, diopside is blue (Figure 8), then light green, then bright green commonly.

Wollastonite–calcite–diopside rock is usually referred to as blue diopside. Diopside forms idiomorphic isometric crystals, and wollastonite and calcite form xenomorphic grains. Moreover, these two minerals are not as noticeable as colored diopside. The thickness of blue diopside bodies (zones, as well as lenses, veins) is usually less than 10 cm (although it can be up to 0.5 m) [52]. Polishing is difficult due to the fact that diopside, wollastonite and calcite have different hardnesses and perfect cleavage. The resources of ornamental diopside are estimated at one thousand tons. In the past, the stone was mined illegally and used for the production of souvenirs and jewelry. The exceptionally chemically pure blue diopside is widely used as a reference for magnesium and calcium in microprobe studies.

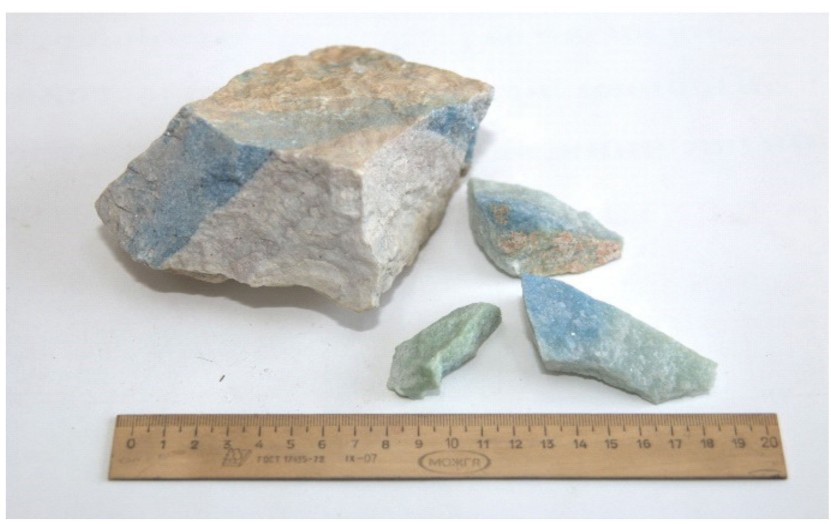

**Figure 8.** A sample of blue diopside. White—wollastonite, pink—foshagite.

### 2.2.7. Zr Mineralization

Several Zr minerals were found in a vein skarn in contact with brucite marbles and troctolites at Belyi stream, including the new mineral dovyrenite [53,54].

Dovyrenite is associated with foshagite, vesuvianite and pyroxene of the series diopside–Al-Ti-diopside ("fassaite"), monticellite, calzirtite, tazheranite, baghdadite, perovskite, apatite, calcite, brucite, melilite, and forsterite. In addition, rare crystals of $Ca_8Zr(Si_2O_7)_2(CO_3)_2(OH)_4$ associated with dovyrenite represent a potential new mineral species. Dovyrenite formed at the regressive stage of early magmatic skarn formation as an alteration product of Zr-bearing and zirconium minerals. Dovyrenite occurs within the central skarn zone in xenoliths (up to a few centimeters in thickness and 10–20 cm in length) composed of dark brown titanian–fassaitic pyroxene. Post-magmatic processes liberated Ti and Zr from pyroxene, leading to newly formed Zr-bearing minerals: perovskite, calzirtite, tazheranite, baghdadite, and dovyrenite. Dovyrenite crystallized in the last phase, frequently forming pseudomorphs after calzirtite [53,54].

## 3. Materials and Methods

### 3.1. Materials

### 3.1.1. Magnesium Silicate Rocks of the Yoko–Dovyren Massif

The chemical composition of magnesium silicate rocks is presented in Table 1.

**Table 1.** Chemical composition of magnesium silicate rocks, wt.%.

| Rock | Basic Oxides | | | | | | | | |
|---|---|---|---|---|---|---|---|---|---|
| | $SiO_2$ | $Al_2O_3$ | $Fe_2O_3$ | FeO | MgO | CaO | $Na_2O$ | $K_2O$ | LOI |
| Dunite | 37.40 | 1.25 | 3.10 | 12.60 | 40.81 | 0.40 | 0.14 | 0.02 | 2.84 |
| Wehrlite | 39.70 | 1.80 | 0.42 | 10.70 | 43.83 | 0.81 | 0.12 | 0.07 | 1.29 |
| Troctolite | 40.60 | 12.00 | 1.11 | 9.45 | 28.60 | 5.57 | 0.57 | 0.04 | 1.33 |
| Dunite sand | 38.40 | 2.10 | 2.93 | 9.95 | 43.20 | 0.46 | 0.05 | 0.03 | 0.98 |

Note: This and the following analyzes were performed by atomic absorption, spectrophotometric, flame photometric, gravimetric, titrimetric methods at the Analytical Center of mineralogical, geochemical and isotope studies at the Geological Institute, SB RAS Ulan-Ude, Russia, analysts A.A. Tsyrenova, T.I. Kazantseva, V.A. Ivanova. LOI—loss on ignition.

Dunite, wehrlite and troctolite differ in their quantitative content of $Al_2O_3$, MgO and CaO. Dunites are 80–97% composed of olivine (chrysolite), clinopyroxene, plagioclase, and secondary serpentine. In wehrlites, in addition to olivine (80–85%), clinopyroxene (diopside) is present. Troctolites are composed of olivine (up to 70%) and plagioclase (anorthite–bitovnite). Dunite sand has the structure of the initial dunites and consists of crystals and fragments of olivine crystals.

### 3.1.2. Portland Cement

Portland cement, without mineral additives, from brand 400 (PC400-D0) of the Timlyuysky cement plant (LLC Timlyuytsement, Russia, Republic of Buryatia) was used for concrete production; the chemical composition of the Portland cement is shown in Table 2.

**Table 2.** Chemical composition of Portland cement, wt.%.

| Basic Oxides | | | | | | | |
|---|---|---|---|---|---|---|---|
| $SiO_2$ | $Al_2O_3$ | MgO | CaO | $Fe_2O_3$ | $Na_2O + K_2O$ | $SO_3$ | LOI |
| 21.60 | 4.91 | 0.91 | 63.11 | 4.08 | 0.78 | 2.55 | 2.43 |

### 3.1.3. Additional Materials

The Portland cement meets the requirements of the Interstate standards GOST 10178-85 "Portland cement and Portland blastfurnace slag cement. Specifications" and GOST 30515-2013 "Cements. General specifications" (Russian Federation).

In the course of the research, crushed stone from granite from the Gornyak quarry (Russia, Republic of Buryatia) was used. A sieve analysis was performed. The partial and total remains of crushed stone in control sieves were determined. The granulometric composition of crushed stone is presented in Table 3. The physical and mechanical properties of crushed stone were studied. The results are shown in Table 4.

**Table 3.** Granulometric composition of the crushed stone.

| Residues on Sieves, % | Sizes of Sieves, mm | | | | Passage |
|---|---|---|---|---|---|
| | 40 | 20 | 10 | 5 | |
| Partial | 0.0 | 21.0 | 58.0 | 17.0 | 4.0 |
| Full | 0.0 | 21.0 | 79.0 | 96.0 | 100.0 |

**Table 4.** Physical and mechanical characteristics of the crushed stone.

| Indicators | Value |
|---|---|
| Volume bulk weight, kg/m$^3$ | 1400.0 |
| True density (specific weight), kg/m$^3$ | 2640.0 |
| Humidity, % | 0.5 |
| Grade by crushing capacity | 1000 |
| Grain content of lamellar (flaky) and needle-shaped grains, % | 24.4 |
| Grain content of weak rocks, % | 7.9 |
| Content of dust-like and clay particles, % | 2.5 |

Table 3 shows the total and partial sieve residues. The partial residue is crushed stone (sand) left on each of the control sieves after the sample has been dispersed. The total residue is the remainder of the crushed stone (sand) that would be on a given sieve if sifting was carried out only through this sieve. It is numerically equal to the sum of the partial residues on a given sieve and all sieves with large openings.

Granite-crushed stone consists of potassium feldspar 60–65%, quartz (25–35%) and biotite (5–10%).

Table 4 shows the grade of crushed stone by crushing. Crushing is the ability of crushed stone to withstand certain loads. Taking into account the magnitude of such loads, a crushed stone grade is formed by crushing. This indicator characterizes the strength of crushed stone and is determined by the weight loss after compression in the cylinder. Crushed stone has a compressive strength of 1100 kg/cm$^2$. After crushing in the cylinder at a load of 200 kN, its weight loss was 11.2%.

Granite-crushed stone of the quarry "Gornyak" meets the requirements of the Interstate standard GOST 8267-93 "Crushed stone and gravel of solid rocks for construction works. Specifications" (Russian Federation) and can be used for the manufacture of concrete grades "100–400".

Quartz–feldspar sand of the quarry "Rechport" (Russia, the Republic of Buryatia) was used as a fine aggregate in concrete manufacture; its granulometric composition is shown in Table 5, and its physical and mechanical characteristics are shown in Table 6.

**Table 5.** Granulometric composition of the sand.

| Residues on Sieves, % | Sizes of Sieves, mm | | | | | | Passage |
|---|---|---|---|---|---|---|---|
| | 5 | 2.5 | 1.25 | 0.63 | 0.315 | 0.16 | |
| Partial | 0.0 | 17.0 | 7.5 | 20.3 | 29.2 | 20.3 | 57 |
| Full | 0.0 | 17.0 | 24.5 | 44.8 | 74.0 | 94.3 | 100.0 |

**Table 6.** Basic properties of the sand.

| Indicators | Value |
|---|---|
| Humidity, % | 2.8 |
| True density (specific weight), kg/m$^3$ | 2650.00 |
| Volume weight (density in the loose-bulk state), kg/m$^3$ | 1700.00 |
| Content of dust-like and clay particles, % | 2.00 |
| Fineness modulus (Mfn) | 2.50 |
| Content of clay in lumps, % | 0.30 |

According to the mineralogical composition, quartz–feldspar sand consists of more than 85% quartz. It contains feldspar, calcite and mica. It also recorded the presence of dusty and clay particles.

According to the fineness modulus and the full residue on sieve no. 0.63, the sand of the quarry "Rechport" is in the group of medium sands, it meets the requirements of the state standard GOST 8736-2014 "Sand for construction works. Specifications" (Russian Federation) and may be suitable for concrete preparation.

The water for concrete mixing met the requirements of the Interstate standard GOST 23732-2011 "Water for concrete and mortars. Specifications" (Russian Federation) in all cases.

*3.2. Methods*

3.2.1. Concrete Sampling

When preparing concrete mixes, the same amount of crushed stone of four types with the largest aggregate grain size of 5–40 mm was used. The ratio of the mass of the sand to the total mass of the aggregates was 0.4. The mobility of the concrete mixes was 1–4 cm with a water–solidification ratio of 0.6. Cement consumption remained constant. Concrete samples were formed in cubes with sizes of $100 \times 100 \times 100$ mm.

After 24 h of hardening under normal conditions with an air temperature of $(20 \pm 5)\,°C$, the samples were removed from the mold and placed in a chamber with a temperature of $(20 \pm 2)\,°C$ and a relative humidity of $(95 \pm 5)\%$. The samples were tested after 28 days of hardening.

The method of forming samples, the conditions and the terms of hardening meet the national standards of the Russian Federation.

All tests were performed with three repetitions. The indicators in the article are the average values of the data obtained.

### 3.2.2. Research Methods

The research methodology included performing chemical analyses, as well as physical and mechanical tests.

The chemical analysis was performed by methods of atomic absorption spectroscopy using a Unicam spectrophotometer, SOLAAR–6M (Thermo Electron, Franklin, MA, USA), with the suitable software and gravimetry using a VSL–200/0,1A electronic scale (Nevskiye Vesy, St. Petersburg, Russia).

Mechanical tests were performed on a PG–100 test hydraulic press (DEG, St. Petersburg, Russia) with a load range of up to 10 t and a plate movement speed of $10 \pm 1$ mm/min.

The crushability of crushed stone was determined by the degree of destruction of grains during compression in the cylinder.

Rubble abrasion was determined by weight loss after tests in a shelf drum with balls of the KP—123 brand (Novoye Delo Production Enterprise, St. Petersburg, Russia). The drum rotates at a speed of 30 rpm. The diameter of the ball was $46.30 \pm 0.35$ mm, weight—$405 \pm 10$ g, surface hardness—50–54 HRc.

The abrasion of concrete was determined by weight loss after four test cycles on the laboratory abrasion circle LKI-4 (LLC RNPO RosPribor, Chelyabinsk, Russia) with an abrasive disk rotational speed of $30 \pm 1$ min$^{-1}$, and a vertical load on the sample of $300 \pm 5$ megaPascals.

Crushed stone resistance to impact was determined using a "PM-A" automatic copra (Rostekhnostroy, Krasnodar, Russia) with a striker mass of 5 kg and a drop height of 50 cm. The test time was 40 strokes.

Tests for frost resistance were carried out in the KM freezer (Mayak LLC, Yoshkar-Ola, Russia) with a working temperature of minus 30 °C. The basic method for determining frost resistance was used. The cycle included freezing in air (in a freezer with a temperature of minus $18 \pm 2$ °C) samples saturated with water, and their subsequent thawing in water with a temperature of ($0 \pm 2$ °C. The time for both freezing and thawing was 3 h.

## 4. Results and Discussion

There are various ways to use the overburden and host rocks of mining enterprises [55,56]. These include the construction industry [57,58], metallurgy [59,60], agriculture [61] and others. However, the main use of mining waste is in the construction industry. Using various technological approaches, a wide range of building materials can be derived from them [62–67], including concrete, where the use of mining waste in both large [68,69] and small [70–73] aggregate forms is promising.

Magnesium-containing rocks of the Yoko–Dovyren massif were studied to determine the possibility of using them in the production of building materials. Before involving any raw material in the technological cycle, it is necessary to assess its quality. First of all, its hygienic radiation assessment is performed. It is necessary to avoid exposure to ionizing radiation of natural origin [74–77]. The research is carried out in accordance with the requirements of the Interstate standard GOST 30108-94 "Building materials and elements. Determination of specific activity of natural radioactive nuclei" (the Russian Federation).

As a result of the conducted work, it has been concluded that magnesium silicate rocks belong to the materials of the first class (I). The values of the total specific effective activity of the natural radionuclide Aeff are as follows: dunite—85.69 Bq/kg, verlite—107.89 Bq/kg, troctolite—131.69 Bq/kg, dunite sand—94.45 Bq/kg. The radiation indicators of the rocks do not exceed 370 Bq/kg. According to the requirements of the Interstate standards of the Russian Federation GOST 8267-93 "Crushed stone and gravel of solid rocks for construction works. Specifications" and GOST 8736-2014 "Sand for construction works. Specifications", they can be used for all types of construction work.

It is known that the quality of raw materials has a significant impact on the quality of finished products. Therefore, the physical and mechanical characteristics of magnesium silicate rocks were

studied. Tests of the crushed stone and the sand were carried out using the methods of the Interstate standards of the Russian Federation GOST 8269.0–97 "Crashed stone and gravel from dense rocks and industrial waste for construction works. Physical and mechanical test methods" and GOST 8735–88 "Sand for construction work. Testing methods".

The granulometric composition of crushed stone from magnesium silicate rocks of the Yoko–Dovyren massif was determined by sieving samples on a standard set of sieves and is presented in Table 7.

**Table 7.** Granulometric composition of crushed stone.

| Residues on Sieves, % | Sizes of Sieves, mm | | | | Passage |
|---|---|---|---|---|---|
| | **40** | **20** | **10** | **5** | |
| Dunite | | | | | |
| Partial | 3.1 | 59.0 | 24.2 | 11.5 | 2.2 |
| Full | 3.1 | 62.1 | 86.3 | 97.8 | 100 |
| Wehrlite | | | | | |
| Partial | 3.1 | 58.5 | 24.6 | 11.7 | 2.2 |
| Full | 3.1 | 61.6 | 86.1 | 97.6 | 100 |
| Troctolite | | | | | |
| Partial | 2.9 | 58.6 | 24.8 | 12.1 | 1.6 |
| Full | 2.9 | 61.5 | 86.3 | 98.4 | 100 |

Our analysis of the table data shows that more than 50% of the crushed stone is represented by particles from 20 to 40 mm in size. About 25% of the crushed stone has dimensions from 10 to 20 mm. Particles less than 5 mm in size make up 2.2% of the crushed stone for dunite and wehrlite and 1.6% for troctolite.

The following physical and mechanical parameters of crushed stone from magnesium silicate rocks were determined: crushability, abrasion, and frost resistance. The content of dusty and clay particles, clay in lumps and the presence of foreign clogging impurities were also determined. The stability of crushed stone to environmental influences and the chemical effects of concrete alkalis has been studied. The results obtained are presented in Table 8.

**Table 8.** Physical and mechanical values of crushed stone.

| Indicators | Dunite | Wehrlite | Troctolite |
|---|---|---|---|
| Volume bulk weight, $kg/m^3$ | 1745.0 | 1739.0 | 1728.0 |
| True density (specific weight), $kg/m^3$ | 3004.0 | 3012.0 | 2913.0 |
| Humidity, % | 0.5 | 0.5 | 0.5 |
| Grade by crushing capacity | 1200 | 1200 | 1200 |
| Grade by abrasion capacity | AI | AI | AI |
| Grain content of lamellar (flaky) and needle-shaped grains, % | no | no | no |
| Grain content of weak rocks, % | no | no | no |
| Content of dust-like and clay particles, % | 0.7 | 0.7 | 0.8 |
| Content of clay in lumps, % | no | no | no |
| Mass loss during decay, % | 1.0 | 1.0 | 1.2 |

The grade of crushed stone by crushability, the grade of crushed stone by abrasion and the content of grains in weak rocks are distinguished. The characteristics of the grade in terms of crushing are given in clause 4. The abrasion grade of crushed stone characterizes the wear resistance of the material and is determined by the loss of crushed stone mass after testing in a shelf drum with balls. The selection of grains of weak rocks is carried out as follows: the grains of weak rocks are easily broken by hands

and destroyed by light blows of a hammer. When scratching with a needle, a trace of a steel needle remains on the surface of a weak grain. The crushed stone should not contain grains of weak rocks in an amount of more than 5%.

The compressive strength of crushed stone from dunite is 1365 kg/cm$^2$, wehrlite—1312 kg/cm$^2$, and troctolite—1294 kg/cm$^2$. After crushing in a cylinder at a load of 200 kN, the weight loss of dunite is 9.5%, wehrlite is 9.7%, and troctolite is 10.3%. After abrasion in a shelf drum, the weight loss of dunite is 17.4%, wehrlite is 18.1%, and troctolite is 19.8%.

It has been determined that the magnesium silicate rocks are hard rocks that do not contain grains of weak rocks. According to the number of grains of lamellar and needle forms, they belong to the I group of the crushed stone (up to 10 wt.%). They have a high grade base on their crushing capacity (1200), abrasion (I) and a high specific weight. The crushed stone from these rocks is resistant to the environment and to the chemical effects of alkalis. Using the mineralogical–petrographic method, it was established that dunites, verlites, and troctolites do not contain minerals containing alkali-soluble silica and are inert with respect to alkalis. Crushed stone from these rocks is resistant to all types of decay. Harmful components and impurities in the studied rocks were not detected. They withstand 400 cycles of freezing and thawing and have a brand of frost resistance of F400.

The conducted tests show that the crushed stone from the magnesium silicate rocks in 5–40 mm fractions is of high quality, meets the requirements of GOST 8267-93 and can be used as a concrete aggregate, as well as for roads and other types of construction work.

The granulometric composition of the dunite sand of the Yoko–Dovyren massif, determined by sieving with a standard set of sieves, is shown in Table 9.

**Table 9.** Granulometric composition of sand.

| Residues on Sieves, % | Sizes of Sieves, mm | | | | | | Passage |
|---|---|---|---|---|---|---|---|
| | 5 | 2.5 | 1.25 | 0.63 | 0.315 | 0.16 | |
| Partial | 0.0 | 3.4 | 16.8 | 48.2 | 16.2 | 10.6 | 4.8 |
| Full | 0.0 | 3.4 | 20.2 | 68.4 | 84.6 | 95.2 | 100 |

The sieve analysis has shown that 48.2% of the sand is represented by particles larger than 0.63 mm. According to the fineness modulus (Mfn = 2.72) and the total residue on sieve number 0.63, it belongs to the group of large sands.

The properties of the dunite sand were defined and they are presented in Table 10.

**Table 10.** Basic properties of dunite sand.

| Indicators | Value |
|---|---|
| Humidity, % | 1.0 |
| True density (specific weight), kg/m$^3$ | 3038.0 |
| Volume weight (density in the loose-bulk state), kg/m$^3$ | 1900.0 |
| Content of dust-like and clay particles, % | 3.0 |
| Fineness modulus (Mfn) | 2.72 |
| Content of clay in lumps, % | 0.45 |

As shown by the research, the dunite sand belongs to the group of large sands. It consists of olivine, does not contain layered silicates, amorphous varieties of silicon dioxide, minerals containing sulfur, as well as other harmful components and impurities. A comparison of the color of the alkaline solution, settled over a sample of sand, with the color of the standard solution showed that it does not contain organic impurities. Dunite sand grains have an angular shape with a rough surface. Dunite sand, in its performance, meets the requirements for use as a filler and can be used for all types of construction work.

Since magnesium silicate rocks are of high quality, studies have been conducted on their use as large and small aggregates in the production of concretes. Concrete on granite rubble and quartz sand was taken as a control sample. The mechanical indicators of the concrete were determined, and their dependence on the types of aggregates used was established (Figure 9).

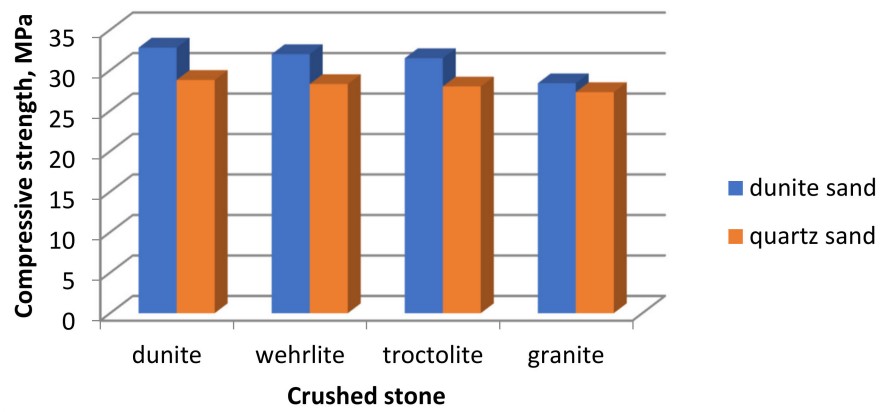

**Figure 9.** Dependence of compressive strength of the concretes on the type of aggregates.

The introduction of the crushed stone from magnesium silicate rocks into the concrete mix leads to an increase in their strength by 2.6–5.5% compared to the control sample. The greatest strength indicators are shown by concretes with the addition of dunite. Moreover, their values depend on the type of sand used. Replacing quartz sand with dunite increases their strength by 12.5–13.9%, depending on the type of crushed stone.

The average density of the resulting concretes (Figure 10) also depends on the type of aggregates. The highest density is found in concretes where dunite crushed stone and dunite sand are used as aggregates.

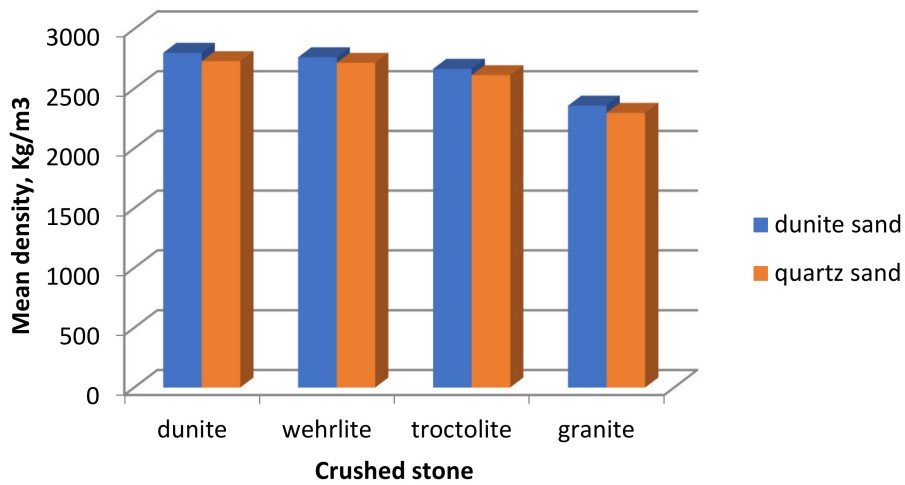

**Figure 10.** Dependence of the average density of the concretes on the type of aggregates.

The frost resistance of concrete with aggregates made of magnesium silicate rocks was determined. The samples withstood 50 freeze–thaw cycles. There was no destruction on their surface. The loss of mass in concrete samples with the addition of dunite aggregates was 1.18%, with the addition of wehrlite the loss was 1.34%, and with troctolite the loss was 1.67%. For the test sample, this figure was 1.83%. Based on the received values, as well as the strength indicators of compression after the completion of tests, the brand of concrete for frost resistance—F50—is defined.

The tests of concretes for abrasion showed that the loss of mass from the samples with magnesium-containing aggregates does not exceed the loss of mass of the control sample (0.63 g/cm$^2$). Based on the classification of the abrasion, the resulting concretes belong to the G1 brand.

The use of magnesium silicate rocks in concrete production allows us to produce heavy concretes (average density 2000–2500 kg/m$^3$), which are used in all load-bearing structures, and, particularly, heavy concretes (average density more than 2500 kg/m$^3$), which are used for the manufacture of special constructions.

The data obtained show that the magnesium-containing rocks are of high quality and can be used in the production of building materials, in particular, concrete (Figure 11).

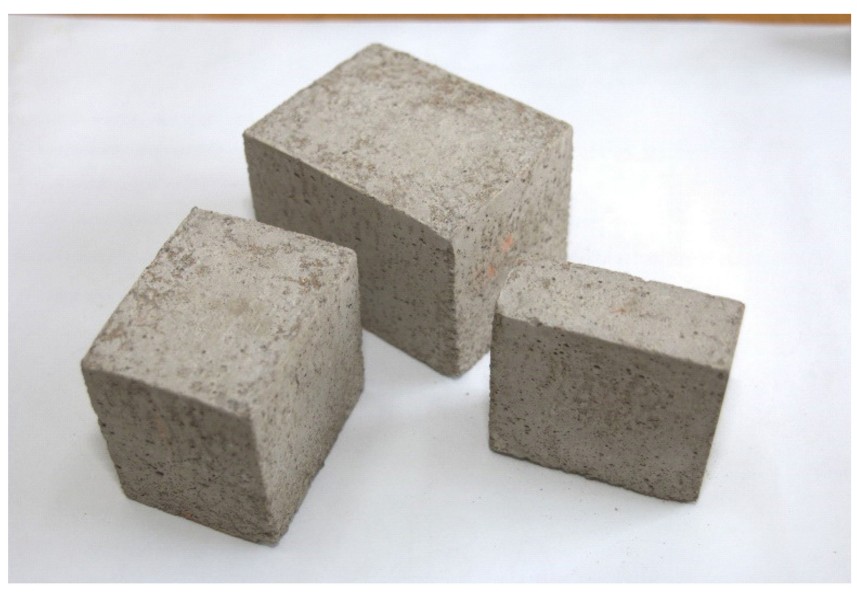

**Figure 11.** Concretes from magnesium-containing rocks of the Yoko–Dovyren massif. Large one have size 20 × 12 × 10 sm$^3$.

When replacing traditional raw materials with them, it is possible to save about 2000 kg of mineral resources while obtaining 1 m$^3$ of concrete.

## 5. Conclusions

Ultramafic–mafic complexes are widely developed in the Earth's crust. They contain deposits of various minerals. The Yoko–Dovyren intrusive in the North Baikal Region, Russia, is considered an example of an intrusive containing diverse mineralization: Ni-Cu, PGE, Cr, Zr, B and blue diopside. Among the magnesium-containing rocks of the Yoko–Dovyren massif are dunites, wehrlites and troctolites, which are moved to dumps during the development of the deposit. The dunites and wehrlite are ultrabasic rocks and the troctolites are basic rocks.

Hygienic radiation assessments of the rocks have shown that they can be used in construction for all types of construction work. The values of the total specific effective activity of the natural radionuclide Aeff are as follows: dunite—85.69 Bq/kg, verlite—107.89 Bq/kg, troctolite—131.69 Bq/kg, dunite sand—94.45 Bq/kg. They do not exceed the normalized values.

The physical and mechanical parameters of the magnesium-containing rocks have been studied and their high quality has been determined. They do not contain lamellar and needle-shaped grains of weak rocks, have a high grade for crushing and a high specific weight, and do not contain harmful components and impurities. The rocks are a promising raw resource for obtaining new types of building materials.

Concretes in large and small aggregate forms were obtained from magnesium-containing rocks. It has been determined that the type of aggregates affects the mechanical characteristics of the concrete.

The strength indicators of the concrete on crushed stone from the studied rocks are 2.6–5.5% higher than that of the control sample. The use of dunite sand increases the strength of the concretes by 12.5–13.9%, depending on the type of crushed stone.

The concrete density also depends on the types of aggregates. Using magnesium-containing rocks, one can obtain heavy concrete for load-bearing structures, and heavy concrete for special structures.

The use of magnesium-containing rocks in the development of deposits, without allowing them to fall into dumps, gives us the opportunity to create clean, environmentally friendly mining enterprises. In addition, replacing traditional raw materials with them will allow us to preserve mineral resources for future generations.

**Author Contributions:** E.V.K. provided the funding, explored the deposit, contributed the raw materials and prepared the paper. L.I.K. conceived the idea, designed the experiments, analyzed the results and prepared the paper. All authors have read and agreed to the published version of the manuscript.

**Funding:** This study was executed within the Geological Institute state task, state registration no. AAAA-A17-117011650012-7, the Baikal Institute of Nature Management state task, state registration no. AAAA-A17-117021310253-8, and funded by Russian Foundation for the Basic Researches according to research project no. 19-05-00337. Our study of the Yoko–Dovyren massif, including field research, sampling, Ni-Cu and PGE mineralization analysis, was carried out partially thanks to the financial support of Russian Science Foundation, grant no. 16-17-10129. The study was conducted using the facilities of the Analytical Center of Mineralogical, Geochemical and Isotope Studies at the Geological Institute, Siberian Branch of Russian Academy of Science, Ulan-Ude, Russia.

**Acknowledgments:** The authors wish to express their sincere thanks to the journal editors and two anonymous reviewers for their constructive comments, which significantly improved the quality of the paper.

**Conflicts of Interest:** The authors declare no conflict of interest.

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
