# Peer review of "Yoko–Dovyren Layered Massif: Composition, Mineralization, Overburden and Dump Rock Utilization"

_minerals, doi:10.3390/min10080682_

Round 1

Reviewer 1 Report

The paper submitted to review is interesting because it presents the possibilities of recucing te amount of mining and processing waste. Owever, I have three basic comment that should be taken into account in the presented material:

  • the publicaton doesn’t refer to the idea of Circular Economy. The content presented in the paper represents this idea ( minimizes the generation of waste),
  • chapter 2 is too expanded. I think it should be significantly shortened,
  • chapter 3.2.1 in mechanical test shows only hydraulic press tests. There is no reference to other very important tests uch as: absorbability, density, frost resistance, abrasion et. al.

Author Response

The authors thank the Reviewers for their attentive and friendly attitude to the paper. I would like to note that in the article the authors set a goal to give a full description of the massif, and as an application - the use of overburden and host rocks. They agree with the comments and respond to them.

1. The first half of the article is significantly reduced - lines 68, 79-82, 85-89, 91-92, 108-110, 130-140, 252-255.
2. Clause 3.2.2. articles expanded. The equipment used for physical and mechanical testing of crushed stone is shown. Lines 406-422.

Reviewer 2 Report

Dear Authors,

The topic of using barren wall rock for construction rather than just dumping it is certainly very relevant. The information given on the geology of the Yoko-Dovyren massif are comprehensive and may be interesting to people interested in geology. While this information covers the whole first half of the paper, the focus of the paper is on the use of magnesium containing rocks for concretes to relief the environment from mining related tailing dumps. The investigation of this core part of the paper is clearly missing substance. The mineral composition of the control material is missing as well as the one of the magnesium containing rock. It cannot be expected from the reader to know the stipulations of the various what the GOST standards cited. So, relevant stipulations would have to be mentioned at least. A respective German standard for concrete aggregates at least requests the material to meet certain requirements with regard to ASR (alkali-silica-reaction), resistance to comminution, to surface attrition, to abrasion, to frost damage etc. None of those parameters were tested. An assumption, that the investigated magnesium containing rocks are suited for "heavy concrete for load-bearing structures, and especially heavy concrete for special structures" needs to be further substantiated with all other tests, certainly also required for concreting aggregates acc. to GOST. In particular the high MgO content of the rock might raise questions with regard to the well known magnesite screed through chemical reaction. 

Further, the purely negative view on impact of mining on nature needs to be reviewed. While in the past mining often led to devastation of environment, nowadays mining and its results may have also a positive influences. Sandy infertile top soils soils in the East German Lusatian lignite district have been replaced with more fertile loess containing soils found in deeper layers in surface mining thus resulting in increased in increased agricultural production afterwards. Precondition is, of cause, a respective legislation and its enforcement. Some statements, such as wall rock exceeding 90% of the extracted ore are to be commented. The figure may be correct for many deposits containing disseminated ore, but certainly not for all those deposits. open pit and underground mining exhibits different values,a s well as disseminated and vein ores, large scale and artesan mining etc.         

Author Response

The authors thank the Reviewers for their attentive and friendly attitude to the paper. I would like to note that in the article the authors set a goal to give a full description of the massif, and as an application - the use of overburden and host rocks. They agree with the comments and respond to them.

Responding to the comments of the Reviewer, I would like to make the following clarifications. All tests of coarse and fine aggregate were performed in full in accordance with the requirements of national standards. The results are presented in tables 7 to 10 and in the text below them. The authors did not submit test methods, as consider that they are standard for all states and there is no need to describe them in the article.
Replies to the comments of the Reviewer are given below.
1. The mineral composition of magnesium-containing rocks in clause 3.1.1 is specified. articles. Lines 345-349.
2. In clause 3.1.3. Mineral compositions of crushed stone and quartz-feldspar sand are presented, which are used as aggregates in the manufacture of control concrete. Lines 363-364, 373-375.
3. In clause 3.2.2. equipment used for physical and mechanical testing of crushed stone and concrete is shown. Lines 406-422
4. Paragraph 4 clarifies the characteristics that were determined when testing crushed stone and sand. Lines 450-452, 478-479, 488-493, 512-520.
5. In paragraph 4. Added physical and mechanical properties of the resulting concrete. Lines 459-463, 467-474, 497-499.
6. The high content of magnesium leads to an uneven change in the volume of cement samples and their cracking, if it is in the form of carbonates. Magnesium in the rocks is in the form of silicates. During the hydration of Portland cement in concrete, the formation of mixed hydrosilicates of calcium, magnesium and iron is possible. They improve the physicomechanical characteristics of cements and do not adversely affect them.
7. A significant amount of waste rocks was specified during the development of deposits associated with ultramafic-mafic massifs. Lines 32-34.

Round 2

Reviewer 2 Report

Dear authors,

I have carefully checked your revisions and can confirm substantial improvements in the quality of presentation of results. Nevertheless. certain important information/explaination is still missing as indicated in the text. 

Author Response

  1. Lines 322, 323 were changed.
  2. Lines 340-344 were added.
  3. Lines 346-350 were added.
  4. Lines 354-358 were added.
  5. Line 309: Research methods and equipment are presented.
  6. Lines 458-469 were added.
